# Antibiotic Resistance in *Campylobacter*: A Systematic Review of South American Isolates

**DOI:** 10.3390/antibiotics12030548

**Published:** 2023-03-09

**Authors:** Ana Beatriz Portes, Pedro Panzenhagen, Anamaria Mota Pereira dos Santos, Carlos Adam Conte Junior

**Affiliations:** 1Center for Food Analysis (NAL), Technological Development Support Laboratory (LADETEC), Federal University of Rio de Janeiro (UFRJ), Cidade Universitária, Rio de Janeiro 21941-598, Brazil; 2Laboratory of Advanced Analysis in Biochemistry and Molecular Biology (LAABBM), Department of Biochemistry, Federal University of Rio de Janeiro (UFRJ), Cidade Universitária, Rio de Janeiro 21941-909, Brazil; 3Graduate Program in Veterinary Hygiene and Technological Processing (PGHIGVET), Faculty of Veterinary Medicine, Fluminense Federal University (UFF), Vital Brazil Filho, Niterói 24220-000, Brazil; 4Analytical and Molecular Laboratorial Center (CLAn), Institute of Chemistry (IQ), Federal University of Rio de Janeiro (UFRJ), Cidade Universitária, Rio de Janeiro 21941-909, Brazil; 5Graduate Program in Food Science (PPGCAL), Institute of Chemistry (IQ), Federal University of Rio de Janeiro (UFRJ), Cidade Universitária, Rio de Janeiro 21941-909, Brazil; 6Graduate Program in Sanitary Surveillance (PPGVS), National Institute of Health Quality Control (INCQS), Oswaldo Cruz Foundation (FIOCRUZ), Rio de Janeiro 21040-900, Brazil; 7Graduate Program in Chemistry (PGQu), Institute of Chemistry (IQ), Federal University of Rio de Janeiro (UFRJ), Cidade Universitária, Rio de Janeiro 21941-909, Brazil

**Keywords:** antimicrobial resistance, food contamination, food of animal origin, meat products, animal husbandry, food-producing animals, antimicrobial susceptibility testing (TSA)

## Abstract

In recent years, *Campylobacter* has become increasingly resistant to antibiotics, especially those first-choice drugs used to treat campylobacteriosis. Studies in South America have reported cases of antibiotic-resistant *Campylobacter* in several countries, mainly in Brazil. To understand the current frequency of antibiotic-resistant *Campylobacter* in humans, farm animals, and food of animal origin in South America, we systematically searched for different studies that have reported *Campylobacter* resistance. The most commonly reported species were *C. jejuni* and *C. coli*. Resistance to ciprofloxacin was found to be ubiquitous in the isolates. Nalidixic acid and tetracycline showed a significantly expressed resistance. Erythromycin, the antibiotic of first choice for the treatment of campylobacteriosis, showed a low rate of resistance in isolates but was detected in almost all countries. The main sources of antibiotic-resistant *Campylobacter* isolates were food of animal origin and farm animals. The results demonstrate that resistant *Campylobacter* isolates are disseminated from multiple sources linked to animal production in South America. The level of resistance that was identified may compromise the treatment of campylobacteriosis in human and animal populations. In this way, we are here showing all South American communities the need for the constant surveillance of *Campylobacter* resistance and the need for the strategic use of antibiotics in animal production. These actions are likely to decrease future difficulties in the treatment of human campylobacteriosis.

## 1. Introduction

*Campylobacter* is a Gram-negative bacterium widely associated with gastroenteritis and enterocolitis in humans worldwide [1]. In the European Union, campylobacteriosis has been the gastrointestinal infection with the highest number of reports in humans since 2005 [2]. In the United States, about 1.3 million cases of the disease are reported annually [2]. The genus *Campylobacter* comprises 32 species, and 9 subspecies have already been described [3]. Among them, the thermophilic species *Campylobacter jejuni* (*C. jejuni*) and *Campylobacter coli* (*C. coli*) are frequently isolated from poultry and pig, respectively [4]. Thermophilic *Campylobacter* is the main cause of bacterial gastroenteritis in humans worldwide [5], mainly *C. jejuni* [6]. This bacterium composes the microbiota of warm-blooded animals and, in most cases, is associated with asymptomatic infections [7].

Meat products, especially chicken, are often contaminated with *C. jejuni,* becoming the main vehicle of human campylobacteriosis through the consumption of undercooked meat [6]. Although evisceration plays an essential role in *C. jejuni* contamination, all steps of the poultry slaughtering process may also have points of contamination. *Campylobacter* contamination in the meat production chain represents a public health hazard and a challenge for health authorities in terms of surveillance, sub-notification, and control. These public health actions are necessary because the clinical manifestation of campylobacteriosis varies from diarrheal cases to more severe diseases such as Crohn’s disease [8], Miller-Fisher syndrome [9], or neurological sequelae such as Guillain–Barré syndrome [10]. The outcome of the disease depends on the immune status of the host, and the use of antibiotics is necessary for the treatment of severe clinical cases in children, the elderly, and immunocompromised individuals [6]. 

Regarding the regulatory framework for *Campylobacter* in food products, especially those of animal origin, the regulations of the U.S., EU, and Australia/New Zealand seem to be the most advanced worldwide. The Food Safety and Inspection Service (FSIS) of the USDA, for example, determines that the maximum acceptable percentage of positive samples can be 15.7% for chicken broiler carcasses, 5.4% for turkey carcasses, 7.7% for chicken parts, and 1.9% for comminuted chicken and comminuted turkey [11]. On the other hand, the EU has a risk assessment framework and a risk assessment model for *Campylobacter* in broilers that sets a maximum of 1000 cfu/g in 50 carcass samples after chilling [12]. Brazil and other South American countries comply with the regulatory framework announced by the EU. Brazil, in particular, has several slaughterhouses that export products to the European Union. However, the inspection service of the Ministry of Agriculture (MAPA) does not have official limits for *Campylobacter* in food of animal origin.

Fluoroquinolones have proved to be first-choice antibiotics in the clinical therapy of campylobacteriosis for many years. However, the widespread use of these drugs in clinical and animal husbandry as growth promoters may have created selective pressure for the emergence of fluoroquinolone-resistant *Campylobacter* in food-producing animals [13,14]. This selection of resistant and multidrug-resistant pathogens represents one of the main challenges for public health actions, mainly in developing countries [15]. Consequently, increasingly frequent multidrug-resistant (MDR) pathogens have emerged worldwide [15]. The World Health Organization (WHO) has included *Campylobacter* on the list of bacteria for which new antibiotics are urgently needed and has classified it as a high-priority pathogen due to the worldwide emergence of strains with a high level of resistance to fluoroquinolones [16]. This increase in resistance to fluoroquinolones has forced the introduction of a new class of antibiotics. Macrolides are currently first-choice antibiotics in the treatment of campylobacteriosis [17]. Within this class, erythromycin has been the most widely used, demonstrating satisfactory therapeutic results. On the other hand, erythromycin resistance levels have been increasing in recent years [18,19], requiring urgent active surveillance. 

*C. jejuni* is a highly adaptable pathogen with several resistance mechanisms, and it requires intense epidemiological surveillance. While it is widely monitored in the developed countries of the European Union and North America, epidemiological surveillance of resistant *Campylobacter* in South America is scarce. This is mainly due to underdiagnosed and underreported cases and the lack of existing studies on this pathogen [20,21,22,23,24]. Given these circumstances, this study aimed to explore the frequency of antibiotic-resistant *Campylobacter* isolates in South American countries among humans, food-producing animals, and food of animal origin, to compile a current distribution of resistant strains and the main sources of contamination.

## 2. Results

### 2.1. The Systematic Review Criteria

The literature search identified a total of 258 articles from the Scopus (*n* = 84), PubMed (*n* = 74), SciELO (*n* = 11), and Embase (*n* = 89) databases. First, duplicates were evaluated by reading the titles and abstracts, which resulted in 197 excluded articles. The remaining 61 were subject to a full-text review. Of these, 12 were eliminated based on the eligibility criteria described in the Materials and Methods. Finally, 49 studies were obtained and subsequently used for the review (Figure 1).

### 2.2. Frequency of Antibiotic-Resistant Campylobacter by South American Countries

Out of 18, nine South American countries had studies showing antibiotic-resistant *Campylobacter* isolates. Brazil presented 24 of the 49 eligible articles, followed by Peru with 6, Chile and Ecuador with 5, Argentina with 3, Paraguay and Trinidad with 2, and Colombia and Uruguay presented only 1 (Appendix A).

In Argentina, *Campylobacter* isolates were resistant to seven antibiotics from five classes. Ciprofloxacin was the antibiotic with the higher rate of resistance, followed by nalidixic acid, tetracycline, ampicillin, enrofloxacin, erythromycin, and gentamicin (Figure 2). In Brazil, *Campylobacter* isolates were resistant to 24 antibiotics from 8 classes. Ciprofloxacin also showed a higher resistance, followed by enrofloxacin, tetracycline, nalidixic acid, ampicillin, erythromycin, amoxicillin, ceftiofur, spectinomycin, gentamicin, azithromycin, colistin, doxycycline, sulfonamide, cephalothin, clarithromycin, streptomycin, norfloxacin, amoxicillin with clavulanic acid, trimethoprim, chloramphenicol, clindamycin, kanamycin, meropenem, and florfenicol. In Chile, *Campylobacter* isolates showed resistance to eight antibiotics from five classes, with aztreonam being the most resistant, followed by ciprofloxacin, tetracycline, ampicillin, amoxicillin and clavulanic acid, erythromycin and chloramphenicol, and spectinomycin and streptomycin. Ecuador showed isolates resistant to seven antibiotics from five classes. These were ciprofloxacin, nalidixic acid, tetracycline, erythromycin, ampicillin, amoxicillin, and gentamicin. In Paraguay, *Campylobacter* resistance was shown to three antibiotic classes and three antibiotics: ciprofloxacin, tetracycline, and erythromycin. Peru had *Campylobacter* isolates resistant to ten antibiotics from seven classes: ciprofloxacin, erythromycin, azithromycin, nalidixic acid, sulfamethoxazole/trimethoprim, ampicillin, tetracycline, cephalothin, ceftriaxone, and chloramphenicol. The *Campylobacter* strains isolated in Trinidad showed resistance to eight antibiotics from seven classes. They were sulfamethoxazole/trimethoprim, ciprofloxacin, streptomycin, oxytetracycline, erythromycin, kanamycin, neomycin, chloramphenicol, and gentamicin. Finally, Uruguay showed strains with resistance to four antibiotics from three classes. All were resistant to clindamycin, telithromycin, nalidixic acid, and tetracycline (Figure 2). A single study in Colombia did not address the phenotypic resistance of isolates, with the study investigating only resistance genes; thus, it was not taken into account [20]. 

### 2.3. The Frequency of Campylobacter Species among Isolates Recovered from South American Eligible Studies

The data showed that *C. jejuni*, *C. coli*, and *C. lari* were the *Campylobacter* species isolated in eligible studies (Table 1). In Argentina, two species were reported, *C. jejuni* (75.6%) and *C. coli* (22.6%), and derived from chicken meat, human, pork, and chicken (Figure 3). In Chile, *C. jejuni* (95.9%) and *C. coli* (4.1%) were reported. This report was from studies with antibiotic-resistant strains of human sources (Figure 3). In Brazil, three species were reported: *C. jejuni* (82.9%), *C. coli* (13.0%), and *C. lari* (0.1%). The studies conducted in this country showed a greater diversity of sources with antibiotic-resistant *Campylobacter*. The sources were chicken meat, chicken, human, environment, pork, and swine (Figure 3). Studies from Colombia and Paraguay displayed one specie each, *C. coli* (100%) and *C. jejuni* (95.4%), respectively (Table 1). Colombia also reported one study with isolates from food (Figure 3). Ecuador and Peru showed the detection of *C. jejuni* (77.1% and 83.5%) and *C. coli* (22.1% and 15.5%), respectively (Table 1). In Ecuador, studies reported isolated strains from chicken, human, chicken meat, cattle, and pork (Figure 3). In Peru, antibiotic-resistant *Campylobacter* was isolated in humans and chicken meat (Figure 3). In Trinidad, *C. coli* (71.7%) and *C. jejuni* (28.2%) were reported (Table 1). These isolated were reported to be from chicken, pork, cattle, sheep, and human (Figure 3). Finally, Paraguay showed studies with isolates from humans and chickens, and in Uruguay, a single study reported *Campylobacter* isolated from a sheep (Figure 3).

## 3. Discussion

*Campylobacter* antibiotic resistance can be developed through spontaneous mutations and the acquisition of resistance determinants can be through natural transformation, transduction, or conjugation [64], according to their different mechanisms of evasion against each antibiotic. Treatment in humans is performed with fluoroquinolones due to their broad spectrum of action, and efficacy against both Gram-negative and Gram-positive bacteria [25]. In food-producing animals, fluoroquinolones are often used to treat infections and as a feed additive indiscriminately. This systematic review showed that studies investigating antibiotic resistance in *Campylobacter* in South America are limited and underexplored. Nevertheless, the results showed that antibiotic resistance in *Campylobacter* had recently increased with concerns regarding resistance against the drugs used as the first choice to treat human campylobacteriosis. Our data compilation showed high levels of ciprofloxacin resistance in seven South American countries (Argentina, Brazil, Chile, Ecuador, Paraguay, Peru, and Trinidad). Peru reached more than 80% of the resistance rate among all antibiotics (Figure 2). High levels of resistance to fluoroquinolone (75–90%) in clinical *Campylobacter* strains have also been reported in countries on other continents [65,66,67], demonstrating the existence of a global health problem. The fast ability of *Campylobacter* to acquire resistance to fluoroquinolones was demonstrated experimentally with only one or two administrations of these antibiotics [13]. The main target of fluoroquinolones in *Campylobacter* is the enzyme DNA gyrase (Topoisomerase II) [68]. This enzyme comprises two subunits, A and B, respectively, encoded by *gyrA* and *gyrB* genes. Its action consists of the catalysis of the ATP-dependent negative supercoiling of DNA and is involved in DNA replication, recombination, and transcription [69]. When fluoroquinolones bind to these enzymes, a stable complex is formed, trapping the enzymes in DNA, leading to DNA double-stranded breaks and bacterial death [69,70]. In *Campylobacter*, the primary mechanism of the development of fluoroquinolone resistance is a single-point mutation in the quinolone resistance-determining region (QRDR) of *gyrA* [17], leading to the substitution of isoleucine for threonine at codon 86 of the *gyrA* gene, which confers a high-level resistance and inhibits bacterial DNA synthesis [70]. Because ciprofloxacin is one of the first options to treat campylobacteriosis, resistance against this drug may compromise antibiotic therapy, posing a public health risk.

Erythromycin was introduced as a substitute in the human clinical treatment of *Campylobacter* infections due to increased resistance to ciprofloxacin [14]. This drug belongs to the macrolide class and, so far, it is the first choice antibiotic for the treatment of human campylobacteriosis [64,71]. Our results showed that erythromycin-resistant *Campylobacter* was isolated in all South American countries except Uruguay (Figure 2). Although rare, the results indicated that erythromycin-resistant *Campylobacter* is spreading, creating a new warning for the use of this drug in animals and humans. In Brazil, macrolides such as tylosin were widely used both as a feed additive [72] and to prevent clostridiosis in swine production. In parallel, an increase in erythromycin-resistant *Clostridium difficile*, which coexists with *Campylobacter* in the intestinal tract of poultry and pigs [72], was observed in several farms using erythromycin. Therefore, the use of macrolides to prevent clostridiosis may have contributed to the selection of macrolide-resistant *Campylobacter* [72]. Tylosin was recently banned as a food additive in Brazil; however, it is still used to treat and prevent animal infections [71]. Macrolides such as erythromycin and tylosin are bacteriostatic. They act by binding to the P site of the 50S ribosomal subunit and inhibit protein synthesis [71]. *Campylobacter* can evade macrolide binding by accumulating mutations in 23S rRNA at position 2074 or 2075 and through other mechanisms such as an efflux pump and altered membrane permeability [14]. The data showed that *Campylobacter* is poorly resistant to macrolides. However, the main concern with macrolide resistance is the compromised treatment of human infections, as erythromycin is currently the main human antibiotic [71].

Another treatment option for campylobacteriosis is tetracycline, which is an antibiotic with broad activity against Gram-negative and Gram-positive bacteria for human and animal treatment and has been used since 1948. The absence of significant adverse side effects has contributed to its widespread use in the treatment of human and animal infections [73]. However, it did not take long before the first case of tetracycline resistance appeared. In 1953, the first tetracycline-resistant bacterium, *Shigella dysenteriae*, was isolated [74]. Currently, resistance to tetracycline has been reported in several bacteria [75,76,77]. *Campylobacter* is becoming increasingly resistant [78,79]. According to our results, seven out of eight South American countries showed *Campylobacter* resistance to tetracycline. Their resistance rate was the second, third, and even the most frequent in these countries (Figure 2). Tetracycline-resistant *Campylobacter* is related to the *tet(O)* gene that encodes the *tet*(O) protein [73]. This protein protects the ribosome from the inhibitory effect of tetracycline and is usually associated with conjugative plasmids [73,80]. Plasmid-mediated resistance spreads faster than chromosomal resistance, thus increasing the emergence of resistant strains. Several studies have reported the appearance of plasmids conferring tetracycline resistance in *C.jejuni* and *C. coli* [80,81,82]. The presence of *tet(O)* in the conjugative plasmids may explain the high distribution of tetracycline resistance found in our results (Figure 2). Tetracycline is widely used to treat animal and human infections [17,83]. Moreover, it is also used as a prophylactic and growth-promoting agent in food-producing animals [84]. The extensive use of tetracycline is due to its broad-spectrum activity and low cost [85]. However, the indiscriminate use of these drugs may put selective pressure on bacteria, pushing the emergence of antibiotic-resistant bacteria that can be transmitted to humans through environment, food, and agricultural workers by direct contact [84].

We observed ampicillin resistance in isolates from Argentina, Brazil, and Chile. In addition, ampicillin-resistant *Campylobacter* were also detected in Ecuador and Peru, but they were in a smaller proportion (Figure 2). Ampicillin belongs to the β-lactam class comprising penicillins, cephalosporins, carbapenems, and monobactams, known by the β-lactam ring in their structures [86]. Their action consists of binding to the penicillin-binding proteins of the bacteria cell wall, inhibiting peptidoglycan synthesis and causing cell lysis [87]. Most *Campylobacter* strains are inherently resistant to many beta-lactams, mainly first- and second-generation penicillins and cephalosporins [17]. They also possess mechanisms that potentiate resistance to this class of antibiotics [17]. The production of beta-lactamases (similar to penicillinases) is the most common and essential resistance mechanism [88]. These enzymes can be encoded by the *bla*_OXA-61_ gene, a chromosomal gene present in most *Campylobacter* strains that confers resistances to β-lactams [89] amoxicillin, ampicillin, and ticarcillin [90]. Therefore, the ampicillin-resistant *Campylobacter* exhibited here may be related to the production of β-lactamases. The expression of penicillinase-type β-lactamase in *Campylobacter* can overcome the β-lactamase inhibitors tazobactam, clavulanic acid, and sulbactam [91]. Here, we found resistance to amoxicillin and amoxicillin with clavulanic acid in Brazil, Chile, and Ecuador, which showed low levels of resistance (Figure 2). The mechanisms of resistance to beta-lactam are not yet fully consolidated but are usually related to the presence of the *bla*_OXA-61_ gene [92]. However, some strains harbor the *bla*_OXA-61_ gene and are not resistant to β-lactams, demonstrating that there may be other mechanisms involved that have not been revealed [92]. In addition, other mechanisms have been described as mediators of β-lactams resistance, such as modifications in outer membrane porins and efflux pumps [86]. Most of the studies performed here in South America have not performed a molecular analysis to identify the genes related to beta-lactam resistance, assessing only phenotypic resistance. The absence of molecular analysis made it difficult to understand the mechanisms of resistance to these antibiotics in South American strains. Therefore, in addition to complement phenotypic tests, molecular analysis should be performed in future studies. Interestingly, a high resistance to aztreonam was found in Chile (Figure 2). Aztreonam is a synthetic monocyclic β-lactam antimicrobial agent belonging to the monobactam family [93]; therefore, its action consists of interfering with the biosynthesis of bacterial cell walls [93], showing an excellent efficacy against Gram-negative bacteria and, due to its poor oral absorption, it is administered intramuscularly or intravenously [94]. However, one of the eligible studies raised the debate that aztreonam was not very efficacious against microaerophilic and aerobic bacteria due to its weak binding to penicillin-binding protein sites in these microorganisms [95]. Therefore, it was initially thought that *Campylobacter* could naturally resist aztreonam. Later, *Campylobacter upsaliensis* strains with sensitivity to aztreonam were found, demonstrating that this resistance did not apply to all species [95]. Still, in general, most *Campylobacter* species are expected to show resistance to this drug.

Resistance to gentamicin was infrequently detected in Argentina, Brazil, and Ecuador. This is probably related to the occasional use of this antibiotic in food-producing animals, mainly because the route of administration is intramuscular, making it difficult to use on a large scale [96]. Furthermore, in specific cases of human infections, severe bacteremia may develop, requiring the intravenous administration of aminoglycosides [97]. Generally, *Campylobacter* exhibits a low resistance to gentamicin [98]. In the United States, gentamicin resistance in *Campylobacter* was rare; the first detection was in 2000 from a human sample and subsequently in 2007, isolated from retail chicken [99]. However, since then, gentamicin resistance in *Campylobacter* has been increasing, presenting higher levels of resistance in isolates detected in 2011 [99]. The main mechanisms of aminoglycoside resistance among Gram-positive and Gram-negative bacteria are an enzymatic modification and antibiotic inactivation [99]. In *Campylobacter*, resistance to gentamicin is not very clear; however, several related genes, such as *aacA4*, *aac(6’)-Ie/aph(2’)-Ia* (also called *aacA/aphD* and encoding a bifunctional enzyme), *aph(2”)*-*If*, and *aph(2″)*-*Ig* have been reported in *Campylobacter* [100,101,102,103]. *aph(2″)-Ig* represents the most recently identified gentamicin resistance gene and encodes a phosphotransferase [100]. 

Resistance to sulfamethoxazole with trimethoprim was only detected in Paraguay and Peru (Figure 2). Strains from Peru presented a low rate of resistance; however, in Paraguay, sulfamethoxazole was the primary antibiotic causing resistance in *Campylobacter*. These results are similar to some European studies in which a high rate of resistance was detected [104,105,106]. Resistance to sulfamethoxazole and trimethoprim was long considered to be intrinsic [107]. However, some studies have shown that it can be acquired through mobile genetic elements by a horizontal gene transfer [108], which should be investigated because of the high spread of these elements. 

A single study from Uruguay reported a multidrug-resistant strain of *Campylobacter fetus* (*C. fetus*) isolated from a sheep abortion [109]. It was the only article in which *C. fetus* was detected. This species is found in the intestinal tract of sheep, cattle, and many other species, causing reproductive disease after reaching the uterus via bacteremia [110]. In sheep, *C. fetus* causes late abortions, stillbirths, and the birth of weak lambs [109] and is recognized as a significant cause of abortions in sheep worldwide [110]. In humans, the main suspected route of transmission of *C. fetus* is the consumption of contaminated animal products or contact with farm animals, causing diarrhea, bacteremia, abortion, and perinatal mortality [111]. We found no studies on *C. fetus* isolated from humans in South America. However, samples tested in this single study from Uruguay showed resistance to four classes of antibiotics: quinolone (nalidixic acid), tetracycline (tetracycline), ketolide (telithromycin), and lincosamide (clindamycin), all at a low frequency. Unfortunately, data on antibiotic resistance in *Campylobacter* in this country are scarce, as is information on antibiotic use, making it difficult to better understand the current situation. This is probably because tetracycline resistance is rarely reported in *C. fetus*, but since it is an antibiotic used to treat campylobacteriosis, more attention should be paid to these findings.

Regarding the origin of *Campylobacter* isolates, we collected studies that detected antibiotic-resistant *Campylobacter* in humans, food-producing animals, food of animal origin, and environmental samples. We found a significant rate of antibiotic-resistant *Campylobacter* isolated from food of animal origin and food-producing animals in most South American countries (Figure 3). Five countries (Argentina, Brazil, Colombia, Ecuador, and Peru) detected a high frequency of antibiotic-resistant *Campylobacter* in food samples. Resistant strains in the food chain are of concern due to the high capacity of human infection through the consumption of contaminated food [112]. Although the in vitro culture of *Campylobacter* is difficult, in the environment, these bacteria can survive under adverse conditions such as acid and oxidative stress [113] and in modified atmosphere packaging [114]. Some species, such as *C. jejuni*, can develop biofilms on abiotic surfaces as a survival mechanism to resist different environmental conditions, thus promoting their permanence in the food production chain and reaching the final product [20,23]. Poultry is the main reservoir of *Campylobacter*, which is usually found in contaminated chicken meat [115]. *Campylobacter* can also be found in other matrices, such as pork and beef [116]. Detections and research are extensive in chickens because the body temperature of a chicken presents the optimal growth temperature for *Campylobacter* (42 °C) [117]. The misuse of antibiotics in poultry selects antibiotic-resistant mutants, which can spread throughout the meat production chain [118]. The presence of *Campylobacter* in food demonstrates the critical role of raw meat in the risk of human exposure. Eligible studies from Trinidad, for example, showed more antibiotic-resistant *Campylobacter* isolated from animal sources than from human infections. One of the Trinidad studies discussed the indiscriminate use of antibiotics in poultry farming, which are widely used as growth promoters and therapeutic agents, often without veterinary guidance. This fact explains the high rate of antibiotic resistance in this country [96]. Brazil presented a great diversity of sources contaminated with antibiotic-resistant *Campylobacter* compared to other countries. They were detected in animals, food of animal origin, environment, and humans (Figure 3). The multiple sources of contamination observed may be related to the more significant number of studies available. On the other hand, it may also indicate an alert for the misuse of antibiotics in this country. The most frequent source was food-producing animals; most isolates were derived from chickens. Food of animal origin also showed a high frequency of chicken meat, the most common source and route of *Campylobacter* infection in humans [6]. Brazil has a significant importance in the world market of chicken meat, being the largest exporter in the world and the third largest producer of chicken meat [116]. However, Brazilian authorities do not set standards for this food pathogen and do not have surveillance programs to control and prevent campylobacteriosis, generating underreporting cases [119].

In parallel to the frequency of resistance, we compile the distribution of *Campylobacter* in terms of the frequency of its isolated species. The results showed that Argentina, Brazil, Chile, Ecuador, Paraguay, and Peru had a higher detection of *C*. *jejuni*, except Colombia and Trinidad, which found higher rates or only the detection of *C*. *coli* (Table 1). This high frequency of *C. jejuni* is related to the frequent isolation of this species of chicken [4], a matrix susceptible to contamination and the primary source of human contamination through the consumption of contaminated chicken meat [120]. In the United Kingdom (UK) and the United States, *C. jejuni* is detected in 90% of human illness [121], often due to the consumption of contaminated chicken meat. Consequently, chicken and chicken meat represent the main source of infection for humans [23,120,122]. According to Suzuki and Yamamoto (2009), although *C. jejuni* is the most widespread, the proportion of *C. jejuni* and *C. coli* varies in some countries [123]. For example, we showed that Colombia detected only *C. coli*, and Trinidad exhibited a more significant detection of *C. coli* than *C. jejuni* (Table 1). This information is divergent in some situations, as studies show *C. jejuni* strains with higher levels of resistance than *C. coli* [122], and others that find no significant differences between them [23].

## 4. Materials and Methods

### 4.1. Search Criteria

A systematic literature search was performed in the Scopus, PubMed, Scielo, and Embase databases following the guidelines of the PRISMA group (Preferred Reporting Items for Systematic Reviews and Meta-Analyses) [124] from January to July 2022. The study protocol was publicly registered at the study’s initiation (PROSPERO CRD 42023389096). The main eligibility criteria were studies published in English, Spanish, or Portuguese, with no publication date limit. The string used the following predetermined groups of keywords that were set individually or in combination: 

Search component 1. “Antibiotic resistance” OR “Antimicrobial resistance”.

Search component 2. “*Campylobacter* or *Campylobacter* spp.”.

Search component 3. “Specific country name” OR “South America”.

The string “specific country name” was represented by the following South American countries: Argentina, Aruba, Bolivia, Brazil, Caribbean Netherlands, Chile, Colombia, Curaçao, Ecuador, Falkland Islands, Guiana, French Guiana, Paraguay, Peru, Suriname, Trinidad and Tobago, Uruguay, and Venezuela. The content of the studies’ bibliography were used to search for other relevant studies that met the eligibility criteria.

#### 4.1.1. Inclusion Criteria


Studies must address the detection of antimicrobial resistance in *Campylobacter* spp.Research must be conducted in South American countries.*Campylobacter* isolates must be derived from humans, food-producing animals, or are divergent.The study must report the total number of samples analyzed and the number of *Campylobacter* isolated from them.Confirmatory testing for *Campylobacter* should be addressed with biochemical and/or PCR/sequencing tests.


#### 4.1.2. Exclusion Criteria 


Incomplete books, reviews, and articles.Studies written in a language other than English, Spanish, or Portuguese.Studies in which *Campylobacter* was not detected or was detected in sources other than humans, food-producing animals, or food of animal origin.Studies that did not perform antibiotic susceptibility testing or showed 100% antimicrobial sensibility.


### 4.2. Focus Questions

The following questions were developed according to the Population, Intervention, Comparison, and Outcome (PICO) method: in which countries in South America have cases of antibiotic-resistant *Campylobacter* been detected again? Which antibiotic has the lowest and highest resistance level against *Campylobacter*? Which sources are most related to the detection of resistant *Campylobacter*? Which *Campylobacter* species are most frequent?

### 4.3. Assessment of the Risk of Bias

Possible sources of bias include the inclusion/exclusion criteria of the study, the database chosen, the language, the number of articles, and the type of article selected for this review. Another essential assessment of bias concerns the different methodologies to evaluate antimicrobial susceptibility. Some studies utilized the minimum inhibitory concentration (MIC), while others used the disc diffusion method.

### 4.4. Frequency Calculations

The frequency of antibiotic-resistant *Campylobacter* in each country was calculated by the ratio of strains exhibiting resistance to a specific antibiotic over the sum of all resistant strains as follows:$$\mathrm{AMR}\;\mathrm{Campylobacter}\;\mathrm{frequency}\;\mathrm{by}\;\mathrm{country}=\frac{\left(n\right)\;\mathrm{strains}\;\mathrm{resistant}\;\mathrm{to}\;a\;\mathrm{specific}\;\mathrm{antibiotic}}{\mathrm{Sum}\;\mathrm{of}\;\mathrm{all}\;\mathrm{resistant}\;\mathrm{strains}}$$

The frequency of antibiotic-resistant *Campylobacter* by the isolation source was calculated by the ratio of all antibiotic-resistant *Campylobacter* in each source over the total number of *Campylobacter* isolates as follows:$$\mathrm{AMR}\;\mathrm{Campylobacter}\;\mathrm{frequency}\;\mathrm{by}\;\mathrm{source}=\frac{\left(n\right)\;\mathrm{resistant}\;\mathrm{strains}\;\mathrm{from}\;\mathrm{each}\;\mathrm{source}}{\mathrm{Sum}\;\mathrm{of}\;\mathrm{all}\;\mathrm{isolated}\;\mathrm{strains}}$$

The frequency of *Campylobacter* species was measured considering all isolates regardless of whether they were antibiotic-resistant or not.

## 5. Conclusions

Studies regarding antibiotic resistance in *Campylobacter* isolated from South American countries need to be better explored. The need for more studies and the lack of reporting of human infection cases prevent the realization of a complete picture, making it challenging to analyze the primary sources related to human infection and the incidence of resistance associated with antibiotic misuse in food-producing animals. Our study alerts all communities to the need for a close surveillance, investigation, and controlled use of ciprofloxacin and tetracycline in South American animal production. These actions will decrease the higher frequency of resistance in *Campylobacter* and reduce the hazard of infection by this pathogen for various populations.

## Figures and Tables

**Figure 1 antibiotics-12-00548-f001:**
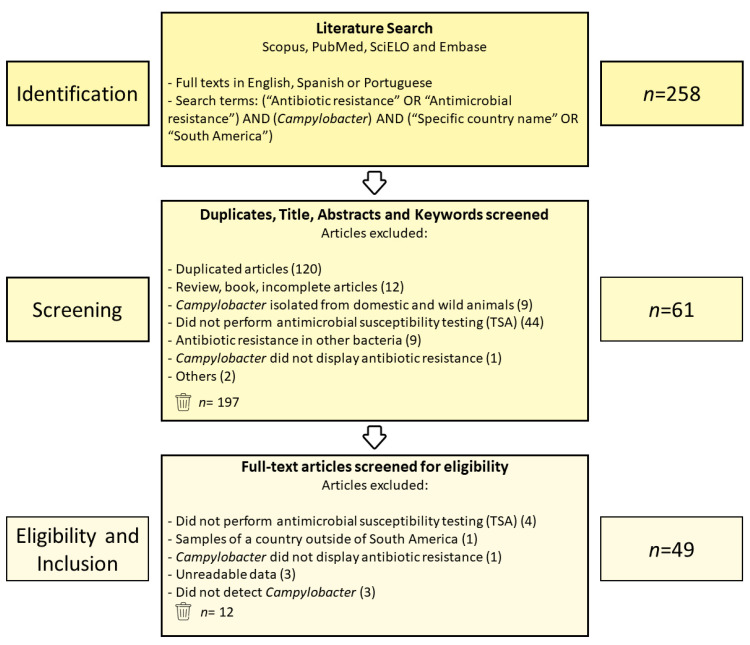
PRISMA flowchart describing the eligibility criteria used in the systematic review process.

**Figure 2 antibiotics-12-00548-f002:**
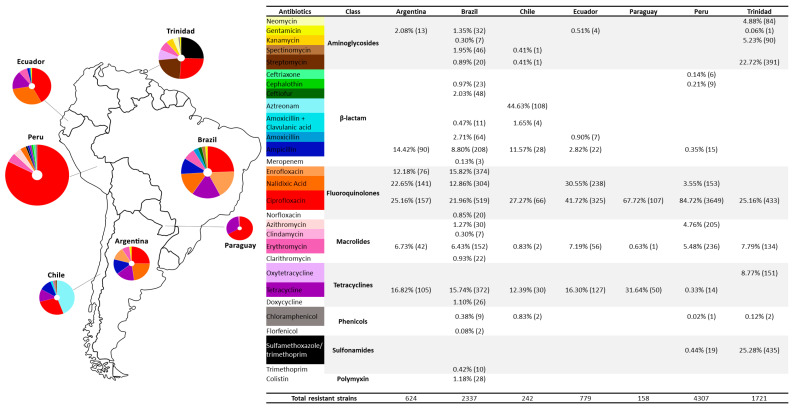
Antibiotic resistance in *Campylobacter* isolates from studies of South American countries. The values in parentheses refer to the amount of *Campylobacter* isolates resistant to the specific antibiotic. Circle diagrams are limited to the ten antibiotics with the highest frequency.

**Figure 3 antibiotics-12-00548-f003:**
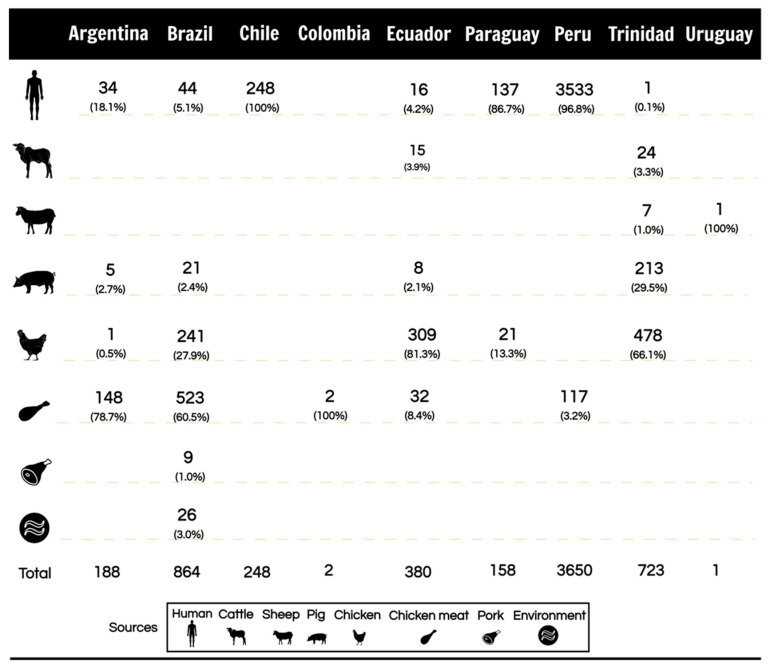
Number and frequency of antibiotic-resistant *Campylobacter* extracted from the eligible studies displayed by the source of isolation and country.

**Table 1 antibiotics-12-00548-t001:** Number and frequency of species among isolated *Campylobacter* strains in South America.

Country	Articles (*n*)	Sample Analyzed	*Campylobacter* Isolated (*n*)	*Campylobacter* Species (*n*)	Frequency (%)	Reference
Argentina	3	50	11	*C. jejuni* (8)	*C. jejuni* (75.6%)	[25]
				*C. coli* (3)	C. coli (22.5%)	
		327	50	*C. jejuni* (48)		[22]
				*C. coli* (2)		
		555	152	*C. jejuni* (105)		[23]
				*C. coli* (43)		
Brazil	22	259	9	*C. jejuni* (5)	*C. jejuni* (82.9%)	[24]
				*C. coli* (3)	*C. coli* (13.0%)	
		92	16	*C. jejuni* (16)	*C. lari* (0.1%)	[25]
		50	34	*C. coli* (14)		[26]
		70	70	*C. jejuni* (69)		[27]
		24	24	*C. jejuni* (22)		[28]
				*C. coli* (1)		
				*C. lari* (1)		
		1	1	*C. jejuni* (1)		[29]
		67	67	*C. jejuni* (67)		[30]
		42	42	*C. jejuni* (14)		[31]
				*C. coli* (25)		
		95	20	*C. jejuni* (18)		[32]
				*C. coli* (2)		
		78	46	*C. jejuni* (39)		[33]
				*C. coli* (7)		
		173	28	*C. jejuni* (28)		[34]
		120	18	*C. jejuni* (5)		[35]
				*C. coli* (13)		
		1070	99	*C. jejuni* (99)		[36]
		159	159	*C. jejuni* (81)		[37]
				*C. coli* (78)		
		54	54	*C. jejuni* (54)		[38]
		116	116	*C. jejuni* (116)		[39]
		442	35	*C. jejuni* (35)		[40]
		141	141	*C. jejuni* (140)		[41]
				*C. coli* (1)		
		50	50	*C. jejuni* (50)		[42]
		515	80	*C. jejuni* (80)		[43]
		2	2	*C. jejuni* (2)		[44]
		48	32	*C. jejuni* (32)		[45]
Chile	5	81	81	*C. jejuni* (69)	*C. jejuni* (95.9%)	[46]
				*C. coli* (12)	*C. coli* (4.1%)	
		50	50	*C. jejuni* (50)		[47]
		108	108	*C. jejuni* (108)		[48]
		73	73	*C. jejuni* (73)		[49]
		350	28	*C. jejuni* (26)		[50]
				*C. coli* (2)		
Colombia	1	2	2	*C. coli* (2)	*C. coli* (100%)	[20]
Ecuador	5	120	50	*C. jejuni* (39)	*C. jejuni* (77.1%)	[51]
				*C. coli* (11)	*C. coli* (22.9%)	
		51	32	*C. jejuni* (22)		[52]
				*C. coli* (10)		
		253	16	*C. jejuni* (13)		[53]
				*C. coli* (3)		
		250	64	*C. jejuni* (49)		[54]
				*C. coli* (15)		
		379	218	*C. jejuni* (170)		[55]
				*C. coli* (48)		
Paraguay	1	150	22	*C. jejuni* (21)	*C. jejuni* (95.4%)	[56]
Peru	6	120	117	*C. coli* (117)	*C. jejuni* (80.4%)	[57]
		189	189	*C. jejuni* (189)	*C. coli* (14.8%)	[58]
		230	19	*C. jejuni* (16)		[59]
				*C. coli* (3)		
		4652	4652	*C. jejuni* (3856)		[60]
				*C. coli* (554)		
		150	106	*C. jejuni* (30)		[61]
				*C. coli* (76)		
		7	7	*C. jejuni* (4)		[62]
				*C. coli* (3)		
Trinidad	1	689	315	*C. jejuni* (89)	*C. jejuni* (28.2%)	[63]
				*C. coli* (226)	*C. coli* (71.7%)	

## Data Availability

Not applicable.

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
