# Peer review of "Antibiotic Resistance in Campylobacter: A Systematic Review of South American Isolates"

_antibiotics, 2023, doi:10.3390/antibiotics12030548_

Round 1

Reviewer 1 Report

  1. The paper titled “Antibiotic Resistance in Campylobacter: A Systematic Review of South America isolates described current frequency of antibiotic-resistant Campylobacter from humans, livestock animals, and food from animal origin in South America using systematic review.
  2. Title is fine and describing the review in efficient manner
  3. Abstract is written good as discussed reason of study, results as well as outcomes. Also add the different stakeholders for which the findings are useful at the end of abstract  
  4. Introduction is written good however, add one paragraph regrading Campylobacter contamination through isolates regarding regulatory framework and rules for the food products especially of animal origin
  5. Material and method section is written is good however, add elaborate inclusion and exclusion criteria along with references for each protocol for the study
  6. Statistical design is missing ?
  7. Results are written in a good way however add the latest references to support findings of the paper
  8. In conclusion major focus should be on findings with practical application
  9. Grammatical mistakes observed on few places so there is need to go through the paper for language and grammatical mistakes

Author Response

Comments and Suggestions for Authors

Reviewer 1

Dear reviewer,

         We are very thankful for all your comments. Your kind consideration were appreciated and very important to improve our manuscript quality. Please find the point-by-point response below:

  1. The paper titled “Antibiotic Resistance in Campylobacter: A Systematic Review of South America isolates” described current frequency of antibiotic-resistant Campylobacter from humans, livestock animals, and food from animal origin in South America using systematic review.

  1. Title is fine and describing the review in efficient manner

  1. Abstract is written well as discussed reason of study, results, and outcomes. Also add the different stakeholders for which the findings are useful at the end of abstract

Response: We changed to: “The level of resistance that was identified may compromise the treatment of campylobacteriosis in human and animal populations.” Please see lines 34-35

  1. Introduction is written good however, add one paragraph regrading Campylobacter contamination through isolates regarding regulatory framework and rules for the food products especially of animal origin

Response: We added a new paragraph in the introduction section regarding this issue. Please see the third paragraph in lines 66-77

  1. Material and method section is written is good however, add elaborate inclusion and exclusion criteria along with references for each protocol for the study

Response: The inclusion and exclusion criteria followed a unique protocol for all search processes regarding the recommendation of the PRISMA protocol. They were not considered individually for each protocol.

  1. Statistical design is missing ?

Response: This study compiled an overall frequency distribution based on the results from collected studies. There was no need for statistical analysis. We added the formulas to improve comprehension of these calculations. Please see 432-437.

  1. Results are written in a good way however add the latest references to support the findings of the paper

Response: We replaced some references we considered too old with more recent ones. Please see reference list.

  1. In conclusion major focus should be on findings with practical application

Response: We reformulated the conclusion section focusing on practical applications. The following sentence refers to the main findings “Our study alerts all communities to the need for close surveillance, investigation, and controlled use of ciprofloxacin and tetracycline in South American animal production. These actions will decrease the higher frequency of resistance in Campylobacter and reduce the hazard of infection by this pathogen for populations.”.

  1. Grammatical mistakes observed on few places so there is need to go through the paper for language and grammatical mistakes

Response: We sent the manuscript to be English grammar checked by a professional service.

Reviewer 2 Report

53. Even though evisceration plays an important role in the contamination by C. jejuni, other steps along the poultry processing steps may also be susceptible to contamination.

82-83, 87-88. Please rephrase. The sentences are vague.

98. The authors may want to state the eligibility criteria as stated in Section 4.1.

Figure 1 is actually a table, not a figure. Consider adding one more column to represent the numbers that represent screening and eligibility instead of adding it at the end of the description. Also, the term “remain” is not suitable Please use a suitable term.

On top of Figure 2, Please add a table to represent the description/data in Section 2.2.

Again, Figure 3 is a table, not a figure. The table is not self-explanatory.

Please state the meaning of the number at the bottom of each country. Add a bottom note to describe the number.

Also, it will be best to represent the true number in addition to the percentage shown.

Table 1. Again, the table is not self-explanatory. Please explain what it means by the numbers at the end of each species. Add a bottom note to explain this.

188. Please use a suitable term instead of ‘reaching’

201-202. Please elaborate more on this. How did the mutation lead to the resistance?

219-220. More elaboration. How does the P site relate to the resistance mechanism?

219. BRASIL 2020. Shouldn’t the reference be numbered?

269. Briefly describe the function of the blaOXA-61 gene and how this relate to the resistance.

359. Use suitable terms instead of food animal (eg: food of animal origin or animal-based food).

Some of the writing in the manuscript suffers from grammatical errors. Please double-check and correct.

Author Response

Comments and Suggestions for Authors

Reviewer 2

Dear reviewer,

         We are very thankful for all your comments. Your kind consideration were appreciated and very important to improve our manuscript quality. Please find the point-by-point response below:

  1. Even though evisceration plays an important role in the contamination by C. jejuni, other steps along the poultry processing steps may also be susceptible to contamination.

Response: We changed to “ Although evisceration plays an essential role in the C. jejuni contamination, all steps of the poultry slaughtering process may also have points of contamination”. Please see lines 56-57

  1. 82-83, 87-88. Please rephrase. The sentences are vague.

Response: We reformulated the entire paragraph to correct the sentences that were vague

  1. The authors may want to state the eligibility criteria as stated in Section 4.1.

Response: We reformulated the sentence as following: “Of these, 12 were eliminated based on the eligibility criteria described in the material and methods”. Please see lines 108-109

  1. Figure 1 is actually a table, not a figure. Consider adding one more column to represent the numbers that represent screening and eligibility instead of adding it at the end of the description. Also, the term “remain” is not suitable Please use a suitable term.

Response: Figure 1 was reformulated to attend to the reviewer's suggestion.

  1. On top of Figure 2, Please add a table to represent the description/data in Section 2.2.

Response: Figure 2 was reformulated to attend to the reviewer's suggestion.

  1. Again, Figure 3 is a table, not a figure. The table is not self-explanatory.

Response: Figure 3 was changed to Table 2 and adjusted to become self-explanatory.

  1. Please state the meaning of the number at the bottom of each country. Add a bottom note to describe the number

Response: These numbers are irrelevant and were removed.

  1. Also, it will be best to represent the true number in addition to the percentage shown.

Response: We added the true number to be displayed beside the percentage.

  1. Table 1. Again, the table is not self-explanatory. Please explain what it means by the numbers at the end of each species. Add a bottom note to explain this.

Response: We changed Table 1 to Table 2 and modified it to turn more self-explanatory.

  1. Please use a suitable term instead of ‘reaching’

Response: We changed to “Treatment in humans is performed with fluoroquinolones due to their broad spectrum of action, and efficacy against both gram-negative and gram-positive bacteria” Please see lines 181-182

  1. 201-202. Please elaborate more on this. How did the mutation lead to the resistance?

Response: We added more comments to elaborate more on this. The sentence added was “The main target of fluoroquinolones in Campylobacter is the enzyme DNA gyrase (Topoisomerase II) [68]. This enzyme comprises two subunits, A and B respectively en-coded by the gyrA and gyrB genes. Its action consists on the catalysis of the ATP-dependent negative supercoiling of DNA and is involved in DNA replication, recombination and transcription [69]. When fluoroquinolones bind to these enzymes, a stable complex is formed, trapping the enzymes in DNA, leading to DNA double-stranded breaks, and bacterial death [69,70]. In Campylobacter the primary mechanism of fluoroquinolone re-sistance development is a single-point mutation in the quinolone resistance-determining region (QRDR) of gyrA [17], leading to substitution of isoleucine for threonine at codon 86 of the gyrA gene, which confers high-level resistance and inhibits bacterial DNA synthesis [70].” Please see lines 195-204

  1. 219-220. More elaboration. How does the P site relate to the resistance mechanism?

Response: We added more comments to elaborate more on this. The sentence added was: “Macrolides such as erythromycin and tylosin are bacteriostatic. They act by binding to the P site of the 50S ribosomal subunit and inhibit protein synthesis [45]. Campylobacter can evade macrolide binding by accumulating mutations in 23S rRNA at position 2074 or 2075 [14] and through other mechanisms such as efflux pump and altered membrane per-meability [76]. The data showed that Campylobacter is poorly resistant to macrolides. However, the main concern with macrolide resistance is the compromised treatment of human infections, as erythromycin is currently the main human antibiotic [14]”. Please see lines 220-227.

  1. BRASIL 2020. Shouldn’t the reference be numbered?

Response: BRASIL 2020 was not a good reference. We changed for a more representative one

  1. Briefly describe the function of the blaOXA-61 gene and how this relate to the resistance.

Response: Response: We added more comments to elaborate more on this. The sentence added was: “ The production of beta-lactamases (similar to penicillinases) is the most common and essential resistance mechanism [88]. These enzymes can be encoded by the blaOXA-61 gene, a chromosomal gene present in most Campylobacter strains that confers resistances to β-lactams [89] amoxicillin, ampicillin, and ticarcillin [90]”. Please see lines 259-262

  1. Use suitable terms instead of food animal (eg: food of animal origin or animal-based food).

Response: We changed to food of animal origin

  1. Some of the writing in the manuscript suffers from grammatical errors. Please double-check and correct.

Response: We sent the manuscript to be English grammar checked by a professional service.